# Lightweight Neural App Control

**Filippos Christianos[1*], Georgios Papoudakis[1*], Thomas Coste[1*], Jianye Hao[1,2], Jun Wang[3], Kun Shao[1†]**

[1]Huawei Noah's Ark Lab, [2]Tianjin University, [3]AI Centre, University College London
`filippos.christianos@huawei.com, georgios.papoudakis1@huawei.com`
`thomas.coste@huawei.com, haojianye@huawei.com, jun.wang@cs.ucl.ac.uk,`
`shaokun2@huawei.com`

## Abstract

This paper introduces a novel mobile phone control architecture, Lightweight Multi-modal App Control (**LiMAC**), for efficient interactions and control across various Android apps. LiMAC takes as input a textual goal and a sequence of past mobile observations, such as screenshots and corresponding UI trees, to generate precise actions. To address the computational constraints inherent to smartphones, we introduce a small Action Transformer (**AcT**) integrated with a fine-tuned vision-language model (VLM) for real-time decision-making and task execution. We evaluate LiMAC on two open-source mobile control datasets, demonstrating the superior performance of our small-form-factor approach against fine-tuned versions of open-source VLMs, such as Florence2 and Qwen2-VL. It also significantly outperforms prompt engineering baselines utilising closed-source foundation models like GPT-4o. More specifically, LiMAC increases the overall action accuracy by up to 19% compared to fine-tuned VLMs, and up to 42% compared to prompt-engineering baselines.

## 1 Introduction

Smartphone application agents, commonly known as app agents, are expanding the potential applications of artificial intelligence to smartphones and other mobile devices. Such agents could allow users to accomplish a range of tasks, from scheduling appointments and sending messages to purchasing items and booking flights, with minimal effort. Fundamentally, app agents observe user instructions and progressively interact with the smartphone's user interface—by clicking, scrolling, inputting text, etc.—to accomplish the task. However, due to the limited computational resources of smartphones, these agents must be optimised for efficiency, employing lightweight models with minimal memory usage and fast processing speeds.

Recent advancements have leveraged foundation models to develop app agents that understand natural language instructions and execute complex user commands within the smartphone's interface (e.g., Rawles et al., 2024; Bai et al., 2024; Wang et al., 2024b;a). While foundation models enable sophisticated capabilities, relying on them for every action introduces significant drawbacks. Their substantial size and computational complexity make them resource-intensive and impractical for constant use on mobile devices. Alternatively, querying server-hosted foundation models, such as GPT-4o or Gemini, for each task can be prohibitively expensive due to the operational costs of running large models, making this approach impractical for everyday applications. For example, a state-of-the-art GPT-4o-based app agent (e.g., Rawles et al., 2024) may require one to two minutes to run and cost approximately \$1.00 per task on average, based on tasks from the evaluated datasets.

To address these limitations, we propose a gated architecture that combines a lightweight transformer network with a small fine-tuned VLM. The task description and the smartphone state are first processed by a compact model ($\sim$500 million parameters) which effectively handles most actions. For actions that require natural language understanding, such as composing a text message or querying a search engine, a VLM is invoked to generate the necessary text. This hybrid approach reduces

---

* First authors with equal contribution.
† Corresponding author.

computational demands and improves responsiveness, resulting in significantly faster execution times—30 times faster, down to 3 seconds per task on average —and improved accuracy.

In the proposed architecture (Lightweight Multi-modal App Control, or LiMAC), the initial processing stage is managed by an Action Transformer (AcT), and is primarily responsible for determining the type of action required to fulfil a user's command. AcT first predicts the action type, such as clicking, inputting text, or scrolling, based on the current state of the smartphone's interface and the task description. For most action types, such as clicks and scrolls, AcT autonomously executes the task. For predicting the targets of the `click` action, we employ a contrastive objective between the outputs of AcT and the embeddings of each user interface (UI) element. The specific approaches for predicting action types and handling click actions are detailed in Sections 3.3 and 3.5, respectively.

However, when the action type predicted by AcT is `input-text` or `open-app`, which necessitate a deeper prior knowledge and understanding of natural language nuances, LiMAC passes the selected action type and user's goal to a fine-tuned VLM to generate the appropriate textual content. This division of labour allows AcT to handle straightforward interactions while leveraging the VLM's advanced capabilities for more complex text generation tasks, ensuring that the system remains both resource-efficient and capable of sophisticated responses. The process of integrating and fine-tuning the VLM in the app agent domain is detailed in Section 3.4.

In summary, the primary contributions of this work are as follows:

- We propose **LiMAC**, an architecture for app agents that balances efficiency and natural language understanding by combining a lightweight transformer with a fine-tuned VLM.

- We also introduce **AcT**, a submodule of LiMAC, which is designed to efficiently predict action types and UI element interactions, featuring a novel contrastive objective for click prediction.

- We fine-tune and evaluate two open-source vision-language models (VLMs) specifically for handling text-based actions. Our fine-tuned VLMs achieve performance comparable to or exceeding GPT-4o methods while only having 2B parameters or less.

- We present experimental results demonstrating that LiMAC improves both task execution time and accuracy—up to **30 times faster** and **40% higher accuracy**—compared to GPT-4o-based and fine-tuned VLM app agents.

## 2 TECHNICAL PRELIMINARIES

### 2.1 PROBLEM FORMULATION

We model phone interaction as a sequential decision-making process. Each task consists of a given goal $g$ that should be completed during an episode. At each timestep $t$ of the episode, the phone's internal state is denoted by $s_t$, while $o_t$ represents an observation of this state, including screen captures and UI element trees. The set of visible UI elements on the screen at timestep $t$ is defined as $\mathcal{I}_t$, with $o_{t,i}$ representing the $i$-th UI element at timestep $t$ where $i \in \mathcal{I}_t$. Each UI element $i$ is represented by three different components: the image that corresponds to the UI element that we denote as $o_{t,i}^{\text{img}}$, the text that corresponds to the UI element $o_{t,i}^{\text{txt}}$, and the related attributes of the UI element, such as whether it is clickable or not, that we denote as $o_{t,i}^{\text{attr}}$. Therefore, the representation of each UI element can be written as:

$$o_{t,i} = [o_{t,i}^{\text{img}}, o_{t,i}^{\text{txt}}, o_{t,i}^{\text{attr}}]. \tag{1}$$

The agent interacts with the phone through actions, denoted as $a_t$ at timestep $t$. Each action is characterised by two components: its type $a_t^{\text{type}} \in \mathcal{A}^{\text{type}}$ (e.g., `click`, `scroll-down`, `input-text`) and its specifications $a_t^{\text{spec}} \in \mathcal{A}^{\text{spec}}$. The specifications vary based on the action type: for clicks, $a_t^{\text{spec}}$ might represent the targeted UI element; for typing actions, it would contain the text to be input. Thus, an action can be represented as the tuple $a_t = (a_t^{\text{type}}, a_t^{\text{spec}})$. This formulation allows for a flexible representation of diverse actions while maintaining a consistent structure.

In this work, the main goal is to learn a model that will maximise action prediction accuracy, which corresponds to correctly predicting both the action type as well as the action specifications. To achieve this, we train AcT, which predicts $a_t^{\text{type}}$. If the predicted action type is `click`, AcT also predicts

the $a_t^{\text{spec}}$ in the form of UI element targets. We focus on click targets because they are among the most difficult and common actions to predict, and AcT's architecture easily accommodates predicting them with a contrastive learning approach (see Section 3.5). For actions that require natural language specifications (e.g., `input-text`), we use a VLM fine-tuned on the same dataset.

## 2.2 SEQUENCE MODELLING WITH TRANSFORMERS

Transformers (Vaswani et al., 2017) have demonstrated exceptional effectiveness in modelling and generating sequential data across a wide range of domains. They excel in various sequence modelling tasks, including those related to language, video processing, and decision-making (Chen et al., 2021). Regardless of the specific application, transformers begin by converting the input into a sequence of vectors. For text, this involves tokenising the input, with each token represented by an embedding vector. In the case of images, the input is typically divided into patches, where each patch is similarly represented by a vector, analogous to the tokenisation process in text. These embeddings, which map tokens or patches to vectors, can either be learned during the model's training or sourced from pre-trained models (e.g., Devlin et al., 2018). The embeddings are fed through several multi-head self-attention layers, which are designed to capture dependencies and contextual relationships between different embeddings in the input sequence. These self-attention mechanisms allow the model to focus on relevant parts of the sequence when processing each embedding, enabling it to handle long-range dependencies more effectively. After passing through multiple layers, each consisting of self-attention and feed-forward components, the final activations from the transformer's last hidden layer are passed through a linear (fully connected) layer. This layer is typically tasked with mapping the learned representations to the output space, whether for classification, prediction, or another specific task. The entire model is trained end-to-end, with backpropagation adjusting both the self-attention layers and the final linear layer to optimise performance on the desired task.

## 3 THE LIGHTWEIGHT MULTI-MODAL APP CONTROL FRAMEWORK

Our methodology processes the user's goal $g$ and the phone's state at time $t$, utilising AcT, to determine the action type $a_t^{\text{type}}$. If the predicted action type is either `input-text` or `open-app`, then $g$, $o_t$, and $a_t^{\text{type}}$ are passed to a fine-tuned VLM, which is responsible for determining the specific action $a_t^{\text{spec}}$. For actions involving clicks, AcT handles the prediction directly but employs a different training objective that contrasts UI element embeddings to determine the most likely interaction target. Accordingly, this section is divided into three parts: predicting the action type, predicting specific actions for text input and app launching, and predicting clicks using our novel approach for interaction with UI elements. The full architecture of LiMAC is presented below.

We refer to our method as *lightweight* because it uses fewer parameters on average during inference than baselines and, as we will show in Section 4.3, has faster inference speeds. The AcT module only has 520M parameters and the additional VLM component is called for less than 15% of actions in our datasets. LiMAC also selects actions more efficiently than a single VLM, as AcT does not require auto-regressive generation. While our approach has a higher memory footprint than solely using VLMs, due to loading both the AcT module and a VLM, its low parameter count remains within the capacity of modern devices (Li et al., 2024b; Laskaridis et al., 2024).

### 3.1 MODEL INPUTS

AcT, the model responsible for predicting the action type (and later the click target, as seen in Section 3.5), is built on top of a typical transformer architecture. However, unlike standard transformers, where tokens represent words or characters, our "tokens" are pretrained embeddings that are mapped to the hidden dimension of the transformer. These tokens represent three key components: the user's goal $g$, the UI elements on the phone's screen $o_{t,i}$, and the possible actions. By using these pretrained embeddings as input, we allow the model to effectively capture the relationships between the user's intent, the current state of the interface, and the set of available actions. We encode each key component (UI elements, actions, and goal) into embeddings that can be processed by the transformer. Below, we describe the encoding process for each type of input.

**Goal:** We encode the user's textual goal $g$ using a sentence encoder, resulting in the embedding $e_g = f_{\text{txt}}(g)$. This embedding captures the user's intent and serves as the first token to the transformer.

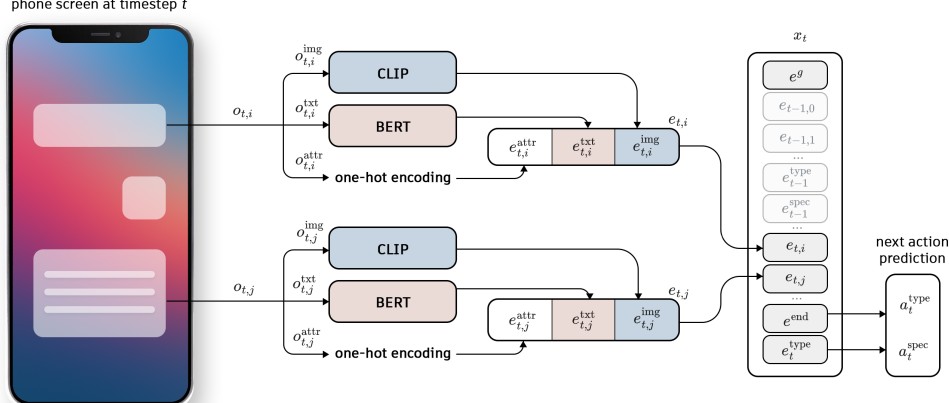

Figure 1: Illustration of AcT. A separate encoding of each UI element into a vector $e_{t,i}$ by using pretrained embedding models. The embeddings are then fed into the sequence of a transformer $x_t$ along with the previous timesteps in that episode. The prediction of the transformer is decoded to produce the next action which consists of $a_t^{\text{type}}$ and $a_t^{\text{spec}}$.

**UI Elements:** The observed representation of each UI element $o_{t,i}$ at time $t$ is transformed into a vector $e_{t,i}^{\text{ui}}$ through several embedding functions. First, the text component is encoded using a sentence encoder (e.g., BERT) $e_{t,i}^{\text{txt}} = f_{\text{txt}}(o_{t,i}^{\text{txt}})$, and the image is encoded using a fine-tuned CLIP visual encoder (Radford et al., 2021) $e_{t,i}^{\text{img}} = f_{\text{img}}(o_{t,i}^{\text{img}})$. Additionally, any other attributes (e.g., clickable, editable, nested) are encoded into $e_{t,i}^{\text{attr}} = f_{\text{attr}}(o_{t,i}^{\text{attr}})$. The final embedding for each UI element is the concatenation of these vectors, $e_{t,i}^{\text{ui}} = [e_{t,i}^{\text{attr}}; e_{t,i}^{\text{txt}}; e_{t,i}^{\text{img}}]$. We fine-tune CLIP using the standard contrastive learning objective (Radford et al., 2021) using the screenshot of the observations and the related UI trees to allow adapting to app control datasets. We also add a positional encoding $p_i \in \mathbb{R}^d$ to represent the order or nesting of UI elements: $e_{t,i}^{\text{ui}} = e_{t,i}^{\text{ui}} + p_i$. This process is illustrated in Figure 1. To adapt the visual encoder $f_{\text{img}}$ to our task, we fine-tune it using our dataset by minimising the InfoNCE loss (Oord et al., 2018), aligning image and text representations of UI elements. Similar methods of representing each UI element as an embedding for the transformer have been suggested by Li et al. (2020); Rawles et al. (2023), with the key distinction that our approach additionally fine-tunes the vision encoder to better adapt it for app control tasks.

**Actions:** Each action is represented using two embeddings: the action type embedding which is mapped to its corresponding learnable embedding $e^{\text{type}}$, and, for actions requiring a specification (e.g., the target of a `click` action), the specification embedding $e^{\text{spec}}$. Depending on the action type, the action specification embedding is computed differently (e.g., sentence embedding for the textual action, learnable embeddings mapped to the UI element's id for click targets, or a special token for empty specifications). Each action contributes two tokens to the transformer's input sequence, clearly separating action types from their parameters.

**Positional Embeddings:** To represent temporal information, we also add a learnable positional encoding $p_t$ for all the embeddings in a timestep.

## 3.2 CONSTRUCTING THE INPUT SEQUENCE

After generating the goal, UI elements, and action embeddings, we organise them into a sequence representing the entire episode. Each episode in the dataset is encoded as a sequence of embeddings $x$, which is fed into the transformer. The sequence starts with the goal embedding $e_g$, followed by the UI element embeddings $e_{0,i}^{\text{ui}}$ at timestep 0. Once all UI elements are encoded, a special end marker $e^{\text{end}}$ is added. The action type $e_0^{\text{type}}$ and specification $e_0^{\text{spec}}$ embeddings for timestep 0 are then appended. This process repeats for each subsequent timestep: encoding UI elements, appending $e^{\text{end}}$, and adding the action embeddings. For an episode with $H$ timesteps, the final sequence is:

$$x = \left[e_g; e_{0,0}^{\text{ui}}; \ldots; e_{0,n}^{\text{ui}}; e^{\text{end}}; e_0^{\text{type}}; e_0^{\text{spec}}; \ldots; e_{H-1,0}^{\text{ui}}; \ldots; e_{H-1,n}^{\text{ui}}; e^{\text{end}}; e_{H-1}^{\text{type}}; e_{H-1}^{\text{spec}}\right]$$

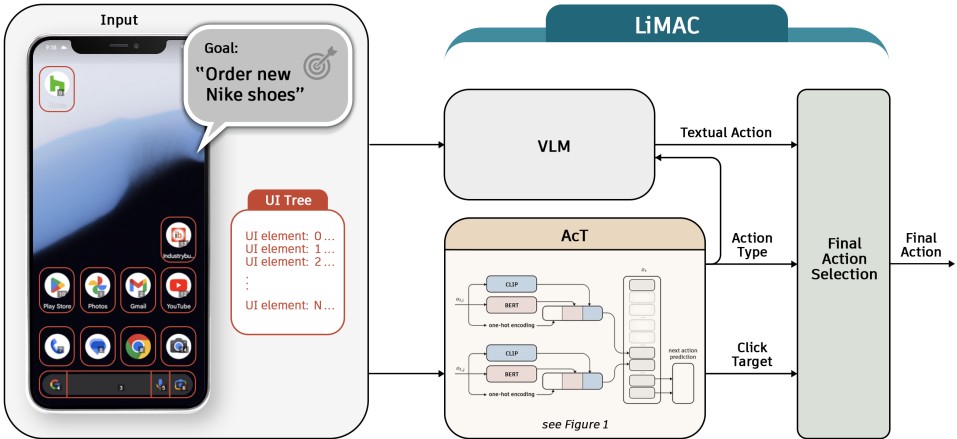

Figure 2: The architecture of LiMAC. The history of observations-actions $\{o_t, a_{t-1}, o_{t-1}..\}$ and goal $g$ are processed to vector $x$ and passed to AcT. The image observation $o_t^{\text{img}}$ with the bounding boxes and the goal $g$ are passed as inputs to the VLM. The VLM is only called if an action that requires text completion is selected, based on the action type output of AcT. The action is finally selected based on the protocol described in Sections 3.3 to 3.5.

During training, the full sequence is fed into the transformer. For inference at timestep $t$, the sequence up to the $t$-th observation is processed, with the hidden state $h_t$ (up to $e^{\text{end}}$) used to predict the action.

### 3.3 ACTION TYPE PREDICTION

In our pipeline, the prediction of the next action begins with determining the action type. Predicting the action type $a_t^{\text{type}}$ can be framed as a classification problem, where we identify a combined eleven distinct action types (see Appendix A), of which a subset are seen in individual datasets used in this work. These action types represent various possible interactions, such as click, open-app, scroll-down, input-text, or other essential commands. We implement the action type prediction with a specialised head. The action type head, denoted as $f_{\text{type}}$, transforms the final hidden state $h_t$ of the transformer (after the $e^{\text{end}}$ token) into a probability distribution over the possible action types, $p(a_t^{\text{type}}|h_t) = f_{\text{type}}(h_t)$. The learning objective for this task is to minimise the cross-entropy loss between the predicted and actual action types. Given a dataset $\mathcal{D}$, the cross-entropy loss for action type prediction is defined as:

$$\mathcal{L}_{\text{type}} = -\mathbf{E}_{a^{\text{type}}, x \in \mathcal{D}} \left[ \log(p(a^{\text{type}}|h)) \right] \tag{2}$$

Here, $h$ represents the transformer's output corresponding to the final hidden state before action prediction, averaged over all observations in the dataset. This loss function ensures that the model is trained to correctly classify the action type based on the sequence of embeddings from previous steps.

### 3.4 LEVERAGING FINE-TUNED VLMS FOR TEXT GENERATION IN ACTION EXECUTION

As described in the previous section, our agent first predicts the action type. Among the eleven action types, two specifically require textual specifications: i) the input-text action, where the specification is the text to be entered into a text box, and ii) the open-app action, where the specification is the name of the application to be opened. For these actions, we rely on fine-tuning a VLM using an app control dataset. The dataset provides action data in a dictionary-like format, such as: {"action-type":"open-app","app-name":"Chrome"}, with one key corresponding to the action type and another to the action specification. The VLM is trained to generate the correct sequence of tokens that corresponds to the successful completion of each action, optimising for the likelihood of generating the proper tokens based on the observation at each timestep.

During inference, after predicting the action type, AcT guides the VLM to start its response with this action type. For instance, if AcT predicts input-text as the action type, the VLM is forced to

begin its response with the token pattern: $\{$`"action-type":"input-text","text":`. The model then completes the specification, producing $a_t^{\text{spec}}$, the textual content needed for the action. The full action selection pipeline is presented in Figure 2.

## 3.5 EFFICIENT CLICK TARGETING USING CONTRASTIVE OBJECTIVES WITH ACT

Having covered how action specifications are generated for textual actions, we now turn to the case of `click` actions, where the specification is the UI element to interact with. To predict the correct UI element for a `click` action, we employ a contrastive learning approach that operates over the entire episode, using cosine similarity and a learnable temperature parameter. Since the number of UI elements varies across timesteps and episodes, a contrastive method is better suited than classification, which can suffer from class imbalance and limitations when handling more UI elements in test episodes than seen during training. Let $h_t^{\text{type}}$ be the transformer's last hidden state up to embedding $e_t^{\text{type}}$, and $f_{\text{target}}$ be an affine transformation that projects the hidden states to an embedding space. Simultaneously, the hidden states of the transformer corresponding to the UI element embeddings, denoted as $h^{\text{ui}}$, are also projected into the same embedding space:

$$q^{\text{type}} = f_{\text{target}}(h_t^{\text{type}}) \quad \text{and} \quad p^{\text{ui}} = f_{\text{target}}(h^{\text{ui}}) \tag{3}$$

Assuming the embedding space lies in $\mathbb{R}^d$, the query embedding $q_t^{\text{type}}$ has dimensions $1 \times D$, while the matrix $p^{\text{ui}}$, representing all UI elements, has dimensions $K \times D$, where $K$ is the total number of UI elements in the episode. The goal is to train the model such that $q_t^{\text{type}}$ aligns closely with the correct UI element's embedding at timestep $t$, using cosine similarity as the alignment measure. To achieve this, we adopt contrastive training techniques with the InfoNCE loss (Oord et al., 2018). We first compute the similarity matrix between the query embedding $q_t^{\text{type}}$ and all UI element embeddings, scaling the similarity by a learnable parameter $\tau$ (e.g., Radford et al., 2021). The scaled cosine similarity matrix is defined as:

$$S = \frac{qp^T}{\|q\| \cdot \|p\|_r} \tau \tag{4}$$

where $\|p\|_r$ is the L2 norm of each row of $p$. For simplicity, we drop the superscripts in this equation. The InfoNCE loss for UI element selection across the episode is computed as:

$$\mathcal{L}_{\text{elem}} = -\mathbf{E}\left[\log \frac{\exp(S_+)}{\sum_{i=1}^{K} \exp(S_i)}\right] \tag{5}$$

Here, $S_+$ is the scaled similarity between the transformer's output and the correct UI element for the `click` action, and $S_i$ represents the similarity between the output and all other UI elements. During inference, for each action requiring a target element, the UI element with the highest similarity is selected. This contrastive approach enables AcT to effectively learn which UI elements to interact with during a `click` action by treating all other UI elements in the episode as negative examples. The use of cosine similarity focuses on the directional alignment of the embeddings, while the learnable temperature $\tau$ adjusts the sharpness of the similarity distribution during training, allowing for more flexible and precise UI element selection.

## 4 EXPERIMENTS

### 4.1 EXPERIMENTAL SETUP

**Datasets:** Our experiments focus on two open-source mobile phone control datasets, AndroidControl (Li et al., 2024a) and Android-in-the-Wild (AitW) (Rawles et al., 2023). Both contain extensive human demonstrations of mobile phone navigation across a wide variety of tasks. In AndroidControl, every episode is defined by a specific goal, accompanied by a sequence of observations and actions. Each observation includes a screenshot from the phone and its corresponding UI tree. Conversely, observations in AitW lack the UI tree. As a result, it is necessary to extract the UI tree using an OCR system that identifies all the UI elements and provides a brief description of each. More details on the goal format, observation space, and action space for each dataset can be found in Appendix A.

**GPT-4o Baselines:** We compare our approach against four prompt-engineering baselines that use GPT-4 to generate actions in the evaluation dataset. First, we evaluate two baselines proposed by

Table 1: Comparison of models in terms of average inference time and overall accuracy on the AitW and AndroidControl datasets. The table presents the size of each model, the average inference time (in seconds, lower is better), and the overall accuracy (higher is better) for both datasets.

| Model | Size ↓ | Avg Inf. Time (s)↓ | Overall ↑ | |
|---|---|---|---|---|
| | | | AitW | AndCtrl |
| SeeAct$_{choice}$ | unk | 9.81 | 37.7 | 29.9 |
| SeeAct$_{ann}$ | unk | 9.76 | 42.5 | 35.5 |
| T3A | unk | 4.87 | 26.9 | 53.1 |
| M3A | unk | 10.64 | 35.6 | 57.5 |
| Florence2 | 820M | 0.50 | 70.8 | 57.0 |
| LiMAC with Florence2 (ours) | +520M | **0.34** | **72.2** | **63.1** |
| Qwen2-VL | 2B | 3.03 | 51.0 | 52.2 |
| LiMAC with Qwen2-VL (ours) | +520M | 0.63 | 70.9 | 62.5 |

Rawles et al. (2024): the text-based **T3A** and the multi-modal **M3A**. In T3A, the observation is represented as a list of UI elements, while M3A includes screenshots of the observation. Additionally, we evaluate two variants of the SeeAct agent (Zheng et al., 2024), adapted for mobile app control tasks (Rawles et al., 2024). Specifically, we assess two SeeAct variants: **SeeAct$_{choice}$** and **SeeAct$_{ann}$**, which use the UI tree text and screenshots of the observations, respectively, to determine the correct action. More details about the prompt engineering baselines are presented in Appendix B.

**Vision Language Models (VLMs):** We fine-tune two VLMs for our experiments. The first, **Florence2** (Xiao et al., 2024), is an 820M-parameter VLM that takes as input an annotated screenshot with numbered bounding boxes, along with the task goal in natural language. Florence2 is trained to maximise the log-likelihood of the correct action tokens from the dataset. Similarly, we fine-tune **Qwen2-VL** (Bai et al., 2023), a 2B-parameter VLM, using LoRA adapters (Hu et al., 2021). Qwen2-VL follows the same pipeline as Florence2, taking the annotated screenshot and goal as inputs, with supervision provided by the correct action. In most of our experiments, these fine-tuned VLMs are tested in conjunction with AcT (forming LiMAC).

## 4.2 Evaluation pipeline

We evaluate on the test set of two datasets, using the same process for all models, with only the observation format and model calling differing. For each timestep, we call the model with the relevant observation format to generate an action. VLMs are trained to return actions in a specific format, while pre-trained models use a detailed prompt with the observation, as in Rawles et al. (2024). One can calculate strict accuracy by directly comparing returned actions to the ground truth. However, in this work we relax this metric for a more practical assessment, where a UI element is deemed correct if its bounding box is within the target element, as described by Li et al. (2024a). For `input-text` actions, correctness is determined by a Jaccard index score of at least 0.5, reflecting the functional equivalence of similar inputs in search bars. We report the *relaxed accuracy* metrics in Tables 1 and 2.

In Section 4.5 we also evaluate models' *action-type* and *click-target* accuracy. Action-type accuracy reflects how well the model predicts the correct type for an action, regardless of the specifications such as the text content or target element. Click-target accuracy measures how accurately the model predicts the correct target for `click` actions when the action type is known. Computing the click-target accuracy requires rerunning a full evaluation over the dataset, where the output of the model is constrained to predict the `click` action and specify the target element. It should be noted that higher overall accuracy can still be achieved with lower action-type and/or click-target accuracy. This is because click-target accuracy is calculated separately, and because not all action types are equally advantageous for overall accuracy. Indeed, as defined in Section 2.1, an action is represented as $a_t = (a_t^{type}, a_t^{spec})$, where both $a_t^{type}$ and $a_t^{spec}$ must be predicted correctly for a successful timestep. Actions which always have a null $a_t^{spec}$, like `wait`, are easier to predict correctly than those which have a complicated $a_t^{spec}$ that may be incorrectly predicted, like `input-text`.

Table 2: Performance comparison of various model configurations using different combinations of modules across the AitW and AndroidControl datasets. Using LiMAC but integrating AcT with baseline methods improves accuracy and reduces inference time (and cost). Not all pairings are shown here for conciseness, the full list can be found in Table 6.

| Framework | Modules Used | | | Avg Inf. Time (s)↓ | Overall↑ | |
|---|---|---|---|---|---|---|
| | Type | Click | Text | | AitW | AndCtrl |
| T3A only | T3A | T3A | T3A | 4.87 | 26.9 | 53.1 |
| LiMAC (ours) | AcT | T3A | T3A | 4.03 | 42.7 | **65.4** |
| LiMAC (ours) | AcT | AcT | T3A | **1.04** | **69.8** | 63.2 |
| M3A only | M3A | M3A | M3A | 10.64 | 35.6 | 57.5 |
| LiMAC (ours) | AcT | M3A | M3A | 8.40 | 52.6 | **66.8** |
| LiMAC (ours) | AcT | AcT | M3A | **1.87** | **70.0** | 62.5 |
| Florence only | Florence2 | Florence2 | Florence2 | 0.50 | 70.8 | 57.0 |
| LiMAC (ours) | AcT | Florence2 | Florence2 | 0.72 | 71.6 | 61.1 |
| LiMAC (ours) | AcT | AcT | Florence2 | **0.34** | **72.2** | **63.1** |
| Qwen only | Qwen2-VL | Qwen2-VL | Qwen2-VL | 3.03 | 51.0 | 52.2 |
| LiMAC (ours) | AcT | Qwen2-VL | Qwen2-VL | 2.64 | 55.7 | 59.1 |
| LiMAC (ours) | AcT | AcT | Qwen2-VL | **0.63** | **70.9** | **62.5** |
| LiMAC (ours) | AcT | M3A | T3A | 7.57 | 52.4 | **67.4** |

## 4.3 MEASURING END-TO-END ACCURACY

In this section, we present the total action accuracy of our method, as well as the baselines. Table 1 present the accuracy for action prediction in AndroidControl and AitW, respectively. In both AitW and AndroidControl, we observe that LiMAC consistently outperforms Florence2, Qwen2-VL, and GPT-4o-based baselines with respect to the action prediction accuracy, demonstrating superior generalisation to the held-out test set. The overall improvement of LiMAC in the accuracy compared to AndroidControl can be attributed to the closer alignment between the training and test sets, as the test set includes the same set of instructions but applied to mobile devices with varying characteristics, such as screen size and Android version. Additionally, we observe a significant performance drop in text-based baselines like T3A and image-text-based models like M3A and SeeAct. The absence of original UI trees in the AitW dataset can explain this decline. Since UI trees must be extracted from images using an annotation tool, inaccuracies are often introduced, which diminishes the performance of models that rely on text-based output conditioning. This underscores a key advantage of LiMAC, which remains robust even when UI trees are imprecise or completely missing (as seen in Table 4), with minimal impact on overall performance.

## 4.4 COMBINING DIFFERENT MODULES

LiMAC is a modular architecture that enables the integration of different modules for tasks such as predicting the action type, identifying the target element in `click` actions, and generating text for `open-app` and `input-text`. In this architecture, we primarily use AcT to predict both the action type and the target element for `click` actions. However, alternative modules can be employed for these predictions as well. In Table 2, we present combinations of different models, excluding SeeAct due to its low overall accuracy, and compare their performance across two datasets.

In the AndroidControl dataset, we observe that using M3A for predicting the target elements in `click` actions improves performance over using AcT alone. This demonstrates that GPT-4o is highly effective at identifying the correct target element when the prompt specifies that the action is `click`. This of courses comes at the cost of calling GPT-4o, which significantly increases the inference time. The highest overall accuracy is achieved when LiMAC is used to predict the action type, M3A is applied for target element prediction, and T3A is used for text generation. In the AitW dataset,

Table 3: Action-type, click-target, and text accuracies across module combinations on the AitW and AndroidControl datasets. LiMAC achieves the best action-type accuracy in both datasets and the best click-target accuracy in AitW, while our fine-tuned Florence2 excels at text prediction.

| Framework | Modules Used | | | Action Type | | Click Target | | Text | |
|---|---|---|---|---|---|---|---|---|---|
| | Type | Click | Text | AitW | AndCtrl | AitW | AndCtrl | AitW | AndCtrl |
| SeeAct only | SeeAct$_{choice}$ | SeeAct$_{choice}$ | SeeAct$_{choice}$ | 67.1 | 66.8 | 36.9 | 48.5 | 69.4 | 67.1 |
| SeeAct only | SeeAct$_{ann}$ | SeeAct$_{ann}$ | SeeAct$_{ann}$ | 68.2 | 66.8 | 44.7 | 55.7 | 66.0 | 61.8 |
| T3A only | T3A | T3A | T3A | 56.2 | 67.7 | 33.5 | 71.1 | 66.5 | **78.4** |
| M3A only | M3A | M3A | M3A | 63.8 | 69.8 | 48.3 | **77.1** | 67.3 | 74.3 |
| Qwen only | Qwen2-VL | Qwen2-VL | Qwen2-VL | 81.7 | 70.7 | 53.2 | 55.2 | 70.5 | 75.7 |
| LiMAC (ours) | AcT | Qwen2-VL | Qwen2-VL | **86.9** | **82.3** | 53.2 | 55.2 | 70.5 | 75.7 |
| LiMAC (ours) | AcT | AcT | Qwen2-VL | **86.9** | **82.3** | **77.4** | 65.4 | 70.5 | 75.7 |
| Florence only | Florence2 | Florence2 | Florence2 | 86.4 | 79.6 | 76.2 | 62.0 | **84.2** | 77.5 |
| LiMAC (ours) | AcT | Florence2 | Florence2 | **86.9** | **82.3** | 76.2 | 62.0 | **84.2** | 77.5 |
| LiMAC (ours) | AcT | AcT | Florence2 | **86.9** | **82.3** | **77.4** | 65.4 | **84.2** | 77.5 |

Table 4: Evaluation of three ablated versions of LiMAC using different types of input, on Android-Control. For actions that require text completion, we use the fine-tuned Florence2.

| | Size | Action Type | Click Target | Overall |
|---|---|---|---|---|
| LiMAC | 520M | 82.3 | 65.4 | **63.1** |
| LiMAC (no CLIP FT) | 520M | 81.9 | 62.3 | 60.0 |
| LiMAC (no img) | 433M | 82.4 | 54.9 | 56.0 |
| LiMAC (no txt) | 410M | **83.2** | **65.7** | 63.0 |

LiMAC combined with Florence for text generation yields the highest accuracy. This outcome is expected, as both M3A and T3A show significantly lower accuracy in this dataset (see Table 1).

## 4.5 ABLATION STUDIES

Table 3 presents the action-type, click-target, and text accuracies for various module combinations across the two datasets. The results show that LiMAC, particularly the AcT, achieves the best performance in action-type prediction. In the AndroidControl dataset, M3A and T3A perform well in click-target and text prediction but struggle with action-type accuracy, and they underperform in the automatically annotated AitW dataset. Overall, AcT within LiMAC excels at click-target predictions while being significantly smaller. Finally, our Florence fine-tune excels at text prediction, significantly outperforming GPT-4o baselines in AitW and remaining competitive in AndroidControl.

Lastly, we present three ablation studies to further explore AcT design choices. A core feature of AcT is its ability to process each UI element as a distinct embedding within the transformer, created by concatenating the image, text, and attribute embeddings of the corresponding UI element. To assess the impact of the image and text modalities, as well as the CLIP fine-tuning on LiMAC's performance, we compare it to three ablated versions: one that excludes the image component, another that omits the UI text in the embedding process, and one that uses the original CLIP for encoding the image embeddings instead of the fine-tuned version. The evaluation metrics for these comparisons in the AndroidControl dataset and using Florence2 for text completion are shown in Table 4. The results demonstrate that removing image embeddings significantly reduces accuracy across all metrics, highlighting the crucial role of visual information in AcT. In contrast, omitting the text embeddings has only a slight effect on performance, suggesting that AcT can function effectively using only screenshots of observations without accessing the UI tree. Additionally, we observe that fine-tuning CLIP (see Section 3.1) is an important factor in improving the overall accuracy of LiMAC.

These findings underscore the importance of visual features and the benefits of fine-tuning pre-trained models like CLIP in our context. The minimal impact of removing text embeddings indicates that LiMAC is robust even when textual information is limited or unavailable, which is advantageous in

scenarios where UI trees are inaccessible or incomplete. Future work could explore integrating other modalities or further optimising the embedding process to enhance performance.

## 5  RELATED WORK ON APP CONTROL

Though graphical user interface (GUI) control mainly started with web-based datasets and foundation model agents (Shi et al., 2017; Liu et al., 2018; Yao et al., 2022a; Deng et al., 2023; Furuta et al., 2023; Gur et al., 2023; Zheng et al., 2024), there has recently been a significant focus on mobile phone control. This can be seen both by the rapid development of Android navigation datasets, environments, and benchmarks (Rawles et al., 2023; 2024; Li et al., 2024a; Chen et al., 2024), and of mobile control agents (Yang et al., 2023; Wang et al., 2024b;a; Wen et al., 2023; Hong et al., 2024; Rawles et al., 2024; Li et al., 2024a; Bai et al., 2024; Wang et al., 2024c). Though many agents are published with their own specific evaluation data, popular datasets such as Android-in-the-Wild (Rawles et al., 2023) or AndroidControl (Li et al., 2024a) are often used as benchmarks. Agents developed for this task can be divided into two clear input types: text-based, using UI accessibility tree or XML information to describe the screen, or image-based. Image-based agents require vision models, which are capable of directly processing image inputs, and are usually backed by VLMs. On the other hand, text-based agents are backed by classical LLMs. Image-based agents also often take a combination of text and image as input to the model. Many mobile control agents propose intricate prompting methods backed by off-the-shelf, often proprietary, LLMs such as GPT-4 (Rawles et al., 2024; Yang et al., 2023; Wang et al., 2024b;a; Wen et al., 2023; Zheng et al., 2024). Although this requires little to no training, it can be both slow and expensive. Moreover, these models cannot be further tailored and trained for specific tasks. As such, another approach is to build mobile control agents around fine-tuned of foundation models on Android control datasets such as AitW or AndroidControl. Firstly, both AitW and AndroidControl present results for a fine-tuned LLM on their dataset, alongside the dataset itself. For example, Li et al. (2024a) train various PaLM 2 (Anil et al., 2023) models on their dataset. However, these models are proprietary and supposedly quite large, with the base PaLM 2 model reported to have over 300B parameters. CogAgent (Hong et al., 2024) also performs fine-tuning on an 18B-large VLM. Bai et al. (2024) propose a different approach, called DigiRL, using RL to train their 1.3B VLM. This achieves strong performance but has limitations such as gathering cost and simulation difficulty, leading to the model only being adept on a small subset of AitW.

## 6  CONCLUSION

In summary, we propose LiMAC, a lightweight framework designed to address app control tasks. LiMAC extracts UI elements from each phone screenshot and encodes them using specialised vision and text modules. These UI element encodings are then passed as embeddings to AcT, which predicts the type and specifications of the next action. AcT focuses on two key aspects of actions: the action type and the target element when the predicted action is `click`. For actions requiring text generation, LiMAC uses a fine-tuned VLM to ensure successful completion. We compare LiMAC against six baselines supported by state-of-the-art foundation models and evaluate them on two open-source datasets. Our results show that LiMAC can outperform the baselines while requiring significantly fewer computational time for both training and inference. This demonstrates that LiMAC is capable of handling task completion on devices with limited computational capabilities.

One of the main limitations of the proposed method is the limited training data. LiMAC is trained on just 13K and 18K episodes for AndroidControl and AitW, respectively. The absence of any pretraining further hinders the model's ability to improve performance on more complex tasks. In the future, we aim to enhance the model's performance by incorporating online learning techniques, such as reinforcement learning. After the initial training stage presented in this work, LiMAC could interact with an Android emulator to generate additional data. By using a suitable reward function, or even leveraging GPT-4 to evaluate the generated trajectories and assign rewards (Bai et al., 2024), we could fine-tune LiMAC to improve the completion rate of tasks. An important focus for future work will be to develop error handling and recovery mechanisms to enable high success rates and robustness in online interactions. Another area of future research could address the safety of such models when handling sensitive data, such as credit card information and personal identifiers. It is essential to design foundation models with robust security protocols to protect against data breaches, especially when interacting with mobile phones containing sensitive information.

## ACKNOWLEDGEMENTS

This work was supported by the National Natural Science Foundation of China (Grant Nos. 62422605, 92370132).

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

# A  DATASET FORMAT

We use AndroidControl and AitW dataset. While we use the full AndroidControl dataset, for AitW we only select a few episodes for each unique instruction, due to the sheer size of the dataset and the repetitive nature of its instructions. The dataset is divided into five categories of tasks, of which we use the 'GoogleApps', 'Install', and 'WebShopping' splits, since the other two contain single-step and Q&A tasks. We process the episodic data present in these datasets into refined, step-wise, datapoints which can be used for training and evaluation. Each datapoint is composed of the high-level goal for the task, an observation of the current screen, and the correct action. Details are given below.

**Goal:** The goal is always a raw text string describing the high-level instruction for the episode to which the datapoint belongs.

**Observation:** The exact form of the observation depends on the type of model it is used for. Text-based approaches such as T3A need a textual observation. For AndroidControl, we use the provided accessibility UI trees, which we further process into a list of UI elements containing information such as the element type, description, and attributes (clickable, editable, selected, etc...). Like in Li et al. (2024a), we filter these to retain only important elements, namely those that contain text or have critical attributes. For AitW, OCR representations of text and icons are given in the dataset, but no comprehensive UI trees are provided. Therefore, to obtain the final representation, each element must be identified and converted using a similar procedure to that for AndroidControl. Vision models such as Qwen2-VL and Florence2 will expect an image-based observation. This observation will consist of the current phone screenshot along with an overlay of the UI element bounding boxes and their index. Finally, some models, such as M3A and ours, use a mixture of observations, both text and image-based. In particular, our model expects a text-based list of UI elements similar to the one described above, as well as a list of cropped images. The list of cropped images corresponds to each of the UI elements in the text-based observation and is used by our model as described in Section 3.1.

**Action:** Action grounding is a crucial part of mobile phone control, so as in previous works (Zheng et al., 2024; Yang et al., 2023; Wang et al., 2024b;a) and both the datasets we use, we define a fixed action space, seen in Table 5. Of this action space, `open-app`, `wait`, and `long-press` do not feature in AitW, while `navigate-home` does not feature in AndroidControl. Information for most actions is sourced directly from the datasets, with only the action name at times varying. The only exceptions to this are the `click` and `long-press` actions, which require a target element, rather than x-y coordinates. For these, we select the best matching candidate from the observation list of UI elements. The action takes a specific JSON format we expect the models to match to facilitate parsing, which is simply a dictionary with the action type and an action specification (see Table 5). An example would be: {`"action-type":"open-app","app-name":"Chrome"`}.

Table 5: Agent action space, along with the relevant datasets.

| Action type | Action specification | AitW | AndCtrl |
|---|---|:---:|:---:|
| open-app | \<app-name\> | ✗ | ✓ |
| click | \<target-element\> | ✓ | ✓ |
| long-press | \<target-element\> | ✗ | ✓ |
| input-text | \<text\> | ✓ | ✓ |
| scroll-{up/down/left/right} | - | ✓ | ✓ |
| navigate-home | - | ✓ | ✗ |
| navigate-back | - | ✓ | ✓ |
| wait | - | ✗ | ✓ |

# B  PROMPT ENGINEERING BASELINES

We evaluate four prompt engineering methods leveraging GPT-4o to generate actions. First, we assess two baselines proposed by Rawles et al. (2024): a text-based method (T3A) and a multi-modal approach (M3A). In both methods, GPT-4o generates a summary of the previous timestep by reflecting on prior actions, the current observation, and previous observations and actions. GPT-4o

then generates a proposed action in a ReAct-like (Yao et al., 2022b) fashion using a detailed prompt that includes task guidelines, action space descriptions, previously generated summaries, and the current observation. In T3A, the observation is represented as a list of UI elements, while in M3A, it also includes two screenshots: one of the original image and another with UI element bounding boxes.

The final two prompt-engineering baselines are SeeAct$_{\text{choice}}$ and SeeAct$_{\text{ann}}$ (Zheng et al., 2024). In both methods, GPT-4o is prompted with the current task and a screenshot from the observation to generate a high-level description of the proposed action. This proposal is then passed to GPT-4o for determining the final action, including both the action type and its specifications in the appropriate format. In SeeAct$_{\text{choice}}$, a multiple-choice list of textual UI element *choices* is appended to the prompt to allow GPT-4o to predict the action specifications, such as the target element in `click` actions. In SeeAct$_{\text{ann}}$, the observation's screenshot is *annotated* with bounding boxes and labels for each UI element. We base our implementation off the SeeAct agent in Rawles et al. (2024), which is adapted to app control tasks.

## C  IMPLEMENTATION DETAILS

AcT is a compact transformer based on GPT-2 architecture. The transformer consists of 24 layers and 16 heads per layer. The hidden dimension of the transformer is 1024. We apply a dropout rate of 0.3 (Srivastava et al., 2014) during training across all layers. The AdamW optimiser (Loshchilov et al., 2017) is used in all experiments, with a learning rate of $3 \times 10^{-4}$ specifically for AcT. The functions $f_{\text{type}}$ and $f_{\text{target}}$ are implemented as two-layer fully connected networks, each with a hidden size of 4096 and a dropout rate of 0.3. We use a batch size of 1 with gradient accumulation being set to 32.

We fine-tune Florence2 for 10 epochs, starting with an initial learning rate of $10^{-6}$, which is gradually reduced to zero during training. The batch size is set to 2, with gradient accumulation configured to 8. For Qwen2-VL, we employ LoRA with a dimensionality of 64, beginning with an initial learning rate of $10^{-4}$, also gradually decreasing to zero throughout training. The batch size for Qwen2-VL is 1, with gradient accumulation similarly set to 8. We fine-tuned Qwen2-VL for 3 epochs.

## D  ADDITIONAL STUDIES

This section presents additional evaluation results for LiMAC.

### D.1  EXTENDED SUCCESS RATE TABLE

In Table 6, we provide a full set of evaluation metrics for the baseline models, as well as for various combinations of LiMAC with other methods. These combinations are used to predict the target element in `click` actions or generate text for specific actions, such as `open-app` and `input-text`. In all the experiments involving LiMAC, AcT is employed to predict the action type, while different combinations of methods are used to predict the action specifications, such as the target element or text generation. This approach allows us to isolate the impact of each combination on performance while maintaining a consistent action type prediction. This table extends the results already presented in Tables 1 to 3 providing a more in-depth understanding of the performance across a range of metrics. This additional breakdown offers a clearer understanding of how LiMAC performs when integrated with other methods, offering insights into the strengths and potential trade-offs of each combination in different scenarios.

### D.2  CONFUSION MATRIX

Figure 3 shows the confusion matrix for action type prediction using LiMAC on the AndroidControl dataset. The results indicate that actions like `open-app` and `input-text` are generally easier to predict compared to other actions. One of the most frequently mispredicted actions is `wait`, which is unsurprising given that it can be challenging, even for humans, to determine when this action is required. Additionally, actions such as `long-press` and `swipe` in any direction are often misclassified, likely due to their relatively low occurrence in the training dataset compared to other actions.

Table 6: Comprehensive table of accuracy results for different modules. All rows which have AcT for the action type module fall under our LiMAC framework.

| Modules Used | | | Action Type | | Click Target | | Text | | Total | |
|---|---|---|---|---|---|---|---|---|---|---|
| Type | Click | Text | AiTW | AndCtr | AiTW | AndCtr | AiTW | AndCtr | AiTW | AndCtr |
| AcT | AcT | Florence2 | 86.9 | 82.3 | 77.4 | 65.4 | 84.2 | 77.5 | 72.2 | 63.1 |
| AcT | Florence2 | Florence2 | 86.9 | 82.3 | 76.2 | 62.0 | 84.2 | 77.5 | 71.6 | 61.1 |
| AcT | AcT | Qwen2-VL | 86.9 | 82.3 | 77.4 | 65.4 | 70.5 | 75.7 | 70.9 | 62.5 |
| AcT | Qwen2-VL | Qwen2-VL | 86.9 | 82.3 | 53.2 | 55.2 | 70.5 | 75.7 | 55.7 | 59.1 |
| AcT | AcT | T3A | 85.3 | 81.7 | 77.6 | 65.4 | 66.5 | 78.4 | 69.8 | 63.2 |
| AcT | T3A | T3A | 85.3 | 81.7 | 33.5 | 71.1 | 66.5 | 78.4 | 42.7 | 65.4 |
| AcT | M3A | T3A | 85.3 | 81.7 | 48.3 | 77.1 | 66.5 | 78.4 | 52.4 | 67.4 |
| AcT | AcT | M3A | 85.3 | 81.7 | 77.6 | 65.4 | 67.3 | 74.3 | 70.0 | 62.5 |
| AcT | T3A | M3A | 85.3 | 81.7 | 33.5 | 71.1 | 67.3 | 74.3 | 43.0 | 64.7 |
| AcT | M3A | M3A | 85.3 | 81.7 | 48.3 | 77.1 | 67.3 | 74.3 | 52.6 | 66.8 |
| AcT | AcT | SeeAct$_{choice}$ | 85.3 | 81.7 | 77.6 | 65.4 | 69.4 | 67.1 | 70.5 | 62.0 |
| AcT | SeeAct$_{choice}$ | SeeAct$_{choice}$ | 85.3 | 81.7 | 36.9 | 48.5 | 69.4 | 67.1 | 45.7 | 53.7 |
| AcT | AcT | SeeAct$_{ann}$ | 85.3 | 81.7 | 77.6 | 65.4 | 66.0 | 61.8 | 70.0 | 61.1 |
| AcT | SeeAct$_{ann}$ | SeeAct$_{ann}$ | 85.3 | 81.7 | 44.7 | 55.7 | 66.0 | 61.8 | 49.2 | 61.6 |
| Florence2 | Florence2 | Florence2 | 86.4 | 79.6 | 76.2 | 62.0 | 84.2 | 77.5 | 70.8 | 57.0 |
| Qwen2-VL | Qwen2-VL | Qwen2-VL | 81.7 | 70.7 | 53.2 | 55.2 | 70.5 | 75.7 | 51.0 | 52.2 |
| T3A | T3A | T3A | 56.2 | 67.7 | 33.5 | 71.1 | 66.5 | 78.4 | 26.9 | 53.1 |
| T3A | M3A | T3A | 56.2 | 67.7 | 48.3 | 77.1 | 66.5 | 78.4 | 30.9 | 55.2 |
| M3A | T3A | T3A | 63.8 | 69.8 | 33.5 | 71.1 | 66.5 | 78.4 | 27.0 | 53.5 |
| M3A | M3A | T3A | 63.8 | 69.8 | 48.3 | 77.1 | 66.5 | 78.4 | 35.8 | 57.7 |
| SeeAct$_{choice}$ | SeeAct$_{choice}$ | SeeAct$_{choice}$ | 67.1 | 66.8 | 36.9 | 48.5 | 69.4 | 67.1 | 29.5 | 38.9 |
| SeeAct$_{ann}$ | SeeAct$_{ann}$ | SeeAct$_{ann}$ | 68.2 | 66.8 | 44.7 | 55.7 | 66.0 | 61.8 | 34.3 | 45.7 |

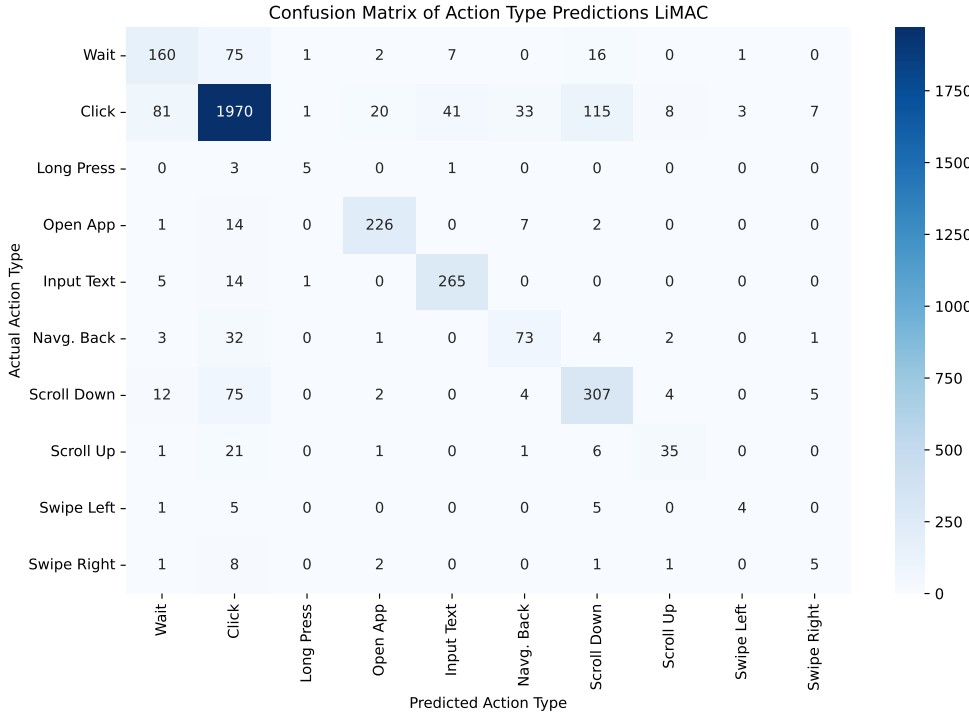

Figure 3: Confusion matrix for action type selection for LiMAC in AndroidControl.

## D.3 FAILURE ANALYSIS

We also examine the failure patterns of LiMAC, using Florence2 as the VLM, across the two datasets studied. Figure 4 displays the frequency of these failures, categorised by the type of failure in predicting either the action type or the action specifications. Specifically, within the action specifications, failures occur in two areas: incorrect prediction of the click target and inaccurate generation of input

text by the VLM. In both datasets, the most common type of failure is misclassification of the action type, closely followed by failures in predicting the click target. These findings underscore the key challenges that research on app control should address.

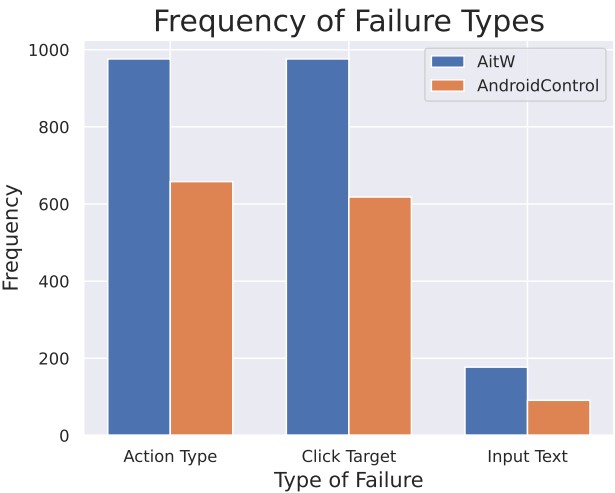

Figure 4: Relative frequency of different types of action prediction errors in the two datasets

## D.4 UI ELEMENTS SCALABILITY

In this section, we assess how the number of UI elements in an observation impacts the success and failure rates of action prediction. Figure 5 displays the number of successful and unsuccessful action predictions made by LiMAC, categorised by the number of UI elements. The results are grouped into bins of ten on the x-axis. The number of UI elements extends up to 150 in AitW and up to 290 in AndroidControl. However, for clarity, only bins containing more than five samples are included in the figures. Overall, the data suggests that the rate of failed action predictions increases slightly as the number of UI elements grows. This trend is expected since accurately predicting the target of `click` actions becomes more challenging with more UI elements present.

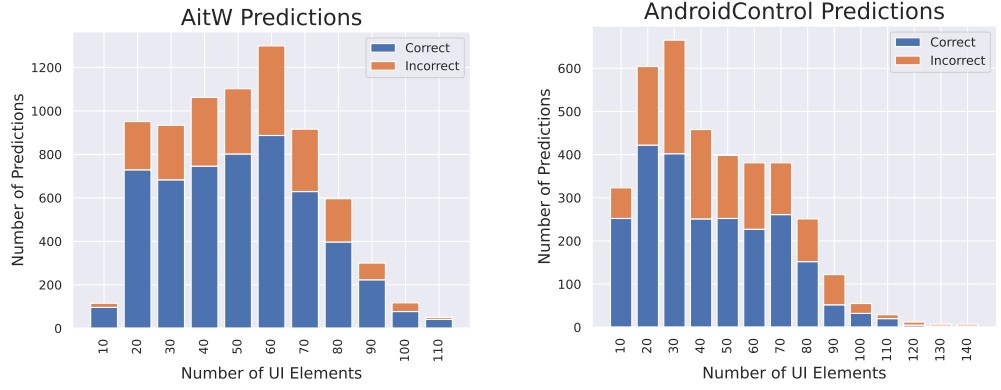

Figure 5: Number of successful and failed prediction of actions with respect to the number of UI elements in the observation, for the two datasets.

# E    CASE STUDIES

Some sample episodes from AndroidControl, including agent predictions, are provided in Figures 6 and 7. These are provided for illustration purposes, as well as to further explain 'relaxed' accuracies and an example failure. Figure 6 presents both an instance of a relaxed target element in the third timestep and a failed `input-text` action in the final timestep. Figure 7 shows a relaxed `input-text` action in the fourth timestep and an otherwise successful episode. Further details are provided in the figure captions.

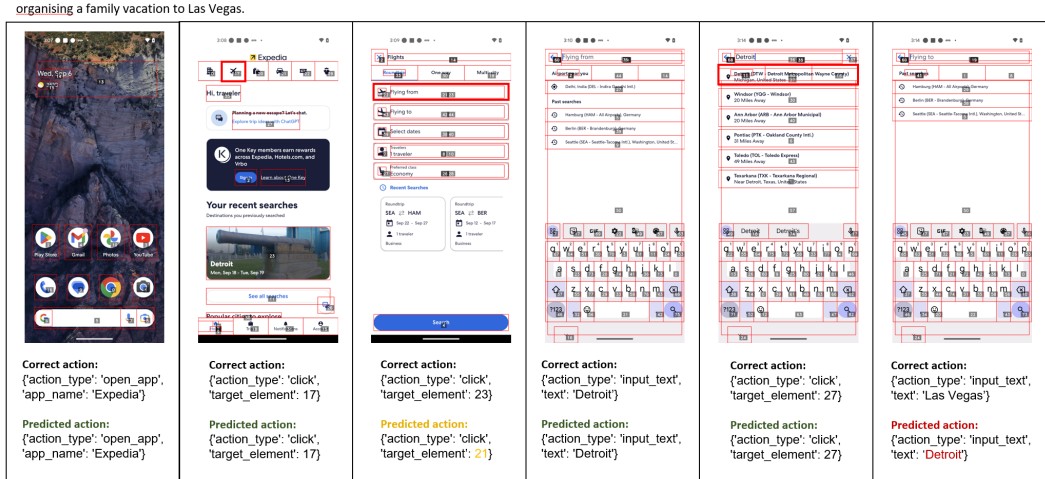

Figure 6: Relaxed target element in yellow (timestep 3) and failed action in red (final timestep). The target element of the `click` in timestep 3 is considered correct under our relaxed accuracy because its bounding box is almost identical to the correct element, and clicking either would have the same effect (opening the text bar). In the final timestep, the agent inputs text 'Detroit' rather than 'Las Vegas', a clear confusion between the origin and destination of the trip stated in the goal, leading to an incorrect prediction.

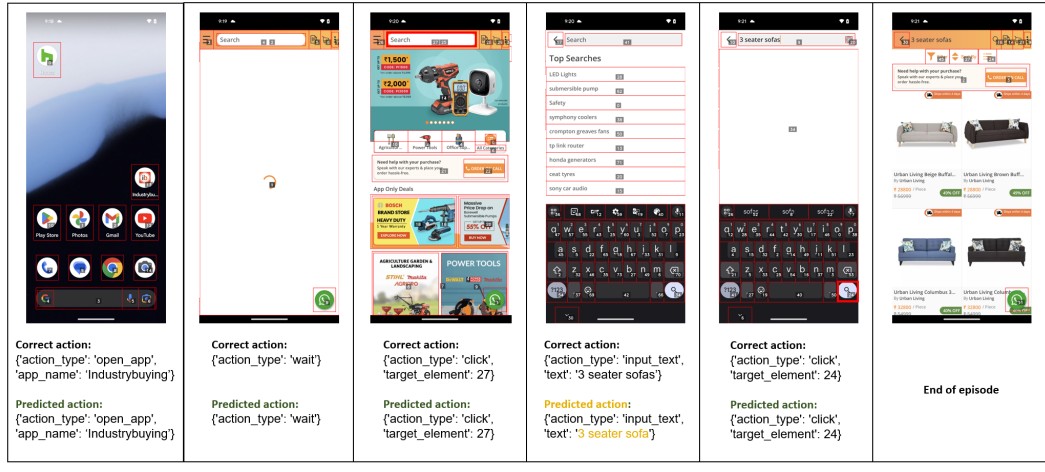

Figure 7: Relaxed `input-text` in yellow (timestep 4) and overall successful episode. Timestep 4 is considered correct under our relaxed `input-text` textual component because it is simply the singular form of the correct text, leading to a Jaccard index greater than 0.5 and presumably the same search results. The episode terminates successfully, with all timesteps being considered correct under our evaluation metrics.