# OpenReview forum: "Lightweight Neural App Control"
_ICLR.cc/2025/Conference — ICLR 2025 Spotlight_

### Official Review · Reviewer_skyy · 2024-10-31

**Soundness:** 3
**Presentation:** 3
**Contribution:** 3
**Rating:** 8
**Confidence:** 4

**Summary:**

This work introduces LiMAC, which is an architecture for training smaller neural networks that could fit on-device for UI control. LiMAC processes UI elements as embeddings, uses a contrasting learning approach for click actions, a gated architecture for selectively invoking a fine-tuning VLM to generate text content, and it shows ablations of the architecture.

The work evaluates several different variances of LiMAC and compares them to multiple baselines, including those using large, proprietary models. In addition to comparing accuracy, they also compare inference time, which is important for on-device applications.

**Strengths:**

The work tackles an under-explored problem of considering compute/inference costs for UI control agents. To get practical agents, the community needs this type of work.
The performance of the LiMAC agent is strong compared to the baselines in the paper. They apply InfoNCE loss in an interesting way for screen understanding and automation. The work includes a large number of baselines, which is appreciated. The authors take off-the-shelve VLMs and fine-tune them for device control, as part of the baselines and main results. And the work include ablation analyses of their results.
The authors explain their methodology well and do rigorous evaluation.

**Weaknesses:**

The results are not that much better compared to the baseline of fine-tuning an off-the-shelf model, particularly for Florence2 as shown in Table 1 (70.8 vs 72.2 for AitW and 57 vs. 63.1 for AndCnrl). It's also hard to tease out if the improved performance is from architectural breakthroughs or rather just from adding more parameters by incorporating the LiMAC network.

On a related note, the claims of the superiority of the proposed architecture would be strengthened with online evaluation by using model-based evaluation on AitW as in DigiRL: https://arxiv.org/abs/2406.11896 or using an online benchmark (e.g., AndroidWorld https://arxiv.org/abs/2405.14573).

The architecture of LiMAC is not particularly novel. While the contrastive loss is interesting, the other parts such as representing UIs using embeddings of UI elements is not novel (past examples: Mapping Natural Language Instructions to Mobile UI Action Sequences: https://arxiv.org/pdf/2307.10088, Android in the Wild: A Large-Scale Dataset for
Android Device Control https://arxiv.org/pdf/2307.10088). While the gated architecture with the VLM is a sensible engineering decision, it is not novel from a research perspective.

**Questions:**

Please see weaknesses.

## Comments
(You do not need to respond to these; they are intended to be helpful)

In future work, you can…
* Do actual on-device implementation. I suspect it may be non-trivial
* You could report mobile-specific performance metrics (battery impact, memory usage, etc.)
* Analyze real-world latency measurements on real phone, which would be compelling

---

> ### Author Response · Authors · 2024-11-19
>
> We would like to thank the reviewer for their review and address their concerns below.
>
> 1. Accuracy Improvement:
>
> While we acknowledge that the improvement in accuracy of LiMAC compared to our fine-tuned Florence does not fully solve the problem, we still believe it represents a meaningful and valuable step forward (and in most cases with large improvements in computational efficiency/speed). We have carefully evaluated LiMAC and provided a thorough analysis of its strengths and weaknesses, which we hope highlights the contribution of our work in advancing the field. Regarding the “more parameters”, while the overall framework does have more parameters, we believe that statement does not paint the full picture, as these parameters do not simply compose a larger network (i.e., not all parameters are called at the same time). For example, if one was to consider action type selection as its own separate task, our 520M AcT model outperformed larger models. Moreover, the parameters of the VLM are only used when the VLM is called, which only occurs for open-app or input-text actions, making up 13.9% of the total actions for AndroidControl and 10.6% for AitW respectively.
>
> 2. Online Evaluation:
>
> We definitely agree that online evaluation is a valuable addition and currently missing from our work, and is an aspect that we aim to address in the future. We believe that fully autonomous, online, and generalisable app agent control, is far from solved [1,2]. Nevertheless, the focus of our study differs from that of DigiRL [3]. In DigiRL, the authors train separate models for different goal categories (General and Webshop) and evaluate on a relatively small subset of goals (the first 96 from the train and test set). In contrast, our work utilises the AndroidControl dataset, which contains more than 13K distinct goals, as well as our subset of AitW, with just under 9K goals. Generalising fine-tuned models in online settings remains an open problem, as highlighted in [1,2] which demonstrate that fine-tuned models perform poorly on a large set of online tasks.
>
> 3. Novelty:
>
> *We have added an acknowledgement of [4] and [5] in Sec 3.1, discussing that they have also explored embedding individual UI elements.*
>
> We would argue that the architecture as a whole is novel even if some of the individual components have been explored before. Indeed, Li et al. [4] (which we thank the reviewer for bringing to our attention) and Rawles et al (AitW) [5] have explored embedding individual UI elements, however we still believe we look deeper into the matter (e.g., using fine-tuned CLIP for the embedding accompanied with ablation studies, using concatenations of embeddings of CLIP and BERT, etc). Furthermore, as the reviewer noted, we use the hidden states of the transformer corresponding to these embeddings for a contrastive learning objective (which to the best of our knowledge has not been explored before). The gated architecture is obviously not a novel idea, and it is pretty natural to the task, but we still believe reporting its capabilities in this context is valuable.
>
> We hope this clarifies our approach and contributions, and we thank the reviewer again for their constructive suggestions. We also appreciate the comments on the on-device implementation.
>
> [1] Chen et al. SPA-Bench: A Comprehensive Benchmark for SmartPhone Agent Evaluation
>
> [2] Zhang et al. LlamaTouch: A Faithful and Scalable Testbed for Mobile UI Task Automation
>
> [3] Bai et al. DigiRL: Training In-The-Wild Device-Control Agents with Autonomous Reinforcement Learning.
>
> [4] Li et. al. Mapping Natural Language Instructions to Mobile UI Action Sequences
>
> [5] Rawles et al Android in the Wild: A Large-Scale Dataset for Android Device Control

---

> > ### Comment · Reviewer_skyy · 2024-11-22
> >
> > Thank you for taking the time to respond and clarify.
> >
> > I agree you have provided careful analysis of LiMAC, which is appreciated. I understand your point regarding the parameters not composing an entire network. While this has clear benefits from a computational efficiency point of view, my concern would be from a memory point of view; specifically regarding how "lightweight" the approach is. Since the focus of this paper is on designing models that could theoretically be useful on device, I believe you would need to keep the larger model loaded and ready to go at all times. Thus for real-world deployment it would effectively be the same as a a larger model from a memory point of view (computationally I agree your approach is faster and more efficient). I understand the actually testing it on-device is outside of the scope of this paper, but this should be considered from a practical point of view. Can you please comment on this?
> >
> > Thank you for the clarification on CLIP and using hidden states of the transformer for a contrastive learning objective. I agree it is novel. Your comment also reminded me of a somewhat related paper, https://arxiv.org/pdf/2012.12350, which uses BERT pre-training task, also using UI elements. Specifically, "Pre-training task #3: Masked VH text prediction", may be of interest for future work.

---

> ### Author Response · Authors · 2024-11-25
>
> We thank the reviewer for their valuable feedback.
>
> *We now include a paragraph with clarifications on the memory footprint and the use of the term “Lightweight” in Sec 3, paragraph 2.*
>
> We acknowledge the concern about memory usage, as our approach does require an additional model to be loaded. While we did not claim a smaller memory footprint, we have now clarified this in Section 3 of the paper.
>
> That being said, there is also some nuance to this discussion, as memory operates differently from compute: its cost is small as long as the device has enough capacity.
> Most modern devices can accommodate transformer-based models up to ~7B parameters [1, 2], though models at the higher end of this range often require aggressive quantization (which impacts performance). Our small VLM + 500M model should be within that limit, even allowing overhead for the OS and open apps. Our use of "lightweight" refers instead to the fact that we i) kept the overall parameter size manageable and likely to fit on a typical modern device (no need for unloading models), leaving some space for the operating system and other apps, ii) are more efficient in our computations (e.g., no auto-regressive generation in AcT), iii) are using less parameters on average for computation (which should translate to less required floating point operations), and iv) simply having faster inference speed. We have now clarified this in Sec 3.
>
> We also thank the reviewer for mentioning ActionBert, as it’s indeed relevant and could be explored in future work.
>
> [1] Li et al. Large Language Model Performance Benchmarking on Mobile Platforms: A Thorough Evaluation.
>
> [2] Laskaridis et al. MELTing point: Mobile Evaluation of Language Transformers.

---

> ### Comment · Reviewer_skyy · 2024-11-26
>
> Thank you for addressing this point. I am raising my score to a 8. I think this paper is well written, spells out its contributions clearly, provides careful analysis with ablations, and makes a new contribution towards actually deploying models on-device.
>
> Note:
>
> > and T3A is used for text generation In the AitW dataset
>
> There is a missing period after generation.

---

> > ### Author Response · Authors · 2024-11-28
> >
> > Thank you for your comment, and thank you for spotting the typo, we will fix it in the next version.

---

### Official Review · Reviewer_j9ZA · 2024-11-02

**Soundness:** 4
**Presentation:** 4
**Contribution:** 3
**Rating:** 8
**Confidence:** 3

**Summary:**

The paper presents a new framework, called LiMAC, for lightweight and efficient Android navigation. Combining text and image embeddings, an action transformer predicts which of ten actions needs to be taken, and, depending on this output, a fine-tuned VLM may be queried to help with more complex tasks, such as text input. A major advantage of this setup is its significantly reduced inference time requirements compared to models such as SeeAct, while attaining higher overall performance on the Android-in-the-Wild and AndroidControl datasets. The authors apply ablation studies to demonstrate the usefulness of CLIP fine-tuning and the image embeddings, as well as the robustness of their setup to the absence of UI tree descriptions. In all, LiMAC is just a 520M-parameter addition, allowing it to run efficiently directly on Android devices.

**Strengths:**

The paper is incredibly clear and well-written. I am not an expert on Android or UI agents, but it was obvious what the contribution was, and why it was important. The small number of parameters in LiMAC make it clear how this advances the ability of users to run advanced Android control models directly on devices. The way the information is encoded is also very thoroughly described. It is somewhat difficult for me to evaluate novelty, but, assuming the related works section does not have any glaring omissions, is a novel approach to solving this problem. The figures are very clear and well-made, and efficiently convey the achitectural design to the reader. Finally, the ablation experiments are well-run, and justify the use of the multiple submodules, as well as demonstrating robustness to a lack of a UI hierarchy description, implying that this method may be applicable out-of-the-box to completely novel Android environments.

**Weaknesses:**

I am not an expert in this area, so it is difficult for me to point out major weaknesses—the paper clearly states a contribution, and is very self-contained. However, there are several minor things that were unclear to me, where the paper might benefit from more detail:

1. The paper mentions that positional encodings are used to represent the nesting of UI elements. How is this done, exactly?
2. It would be good to know what the "ten distinct action types" are—it seems like the same few examples are given at several points in the paper, and only a few are focused on. Are the rest omitted because they're not very interesting and "just work", or are very similar to each other, or something else?
3. In Table 1, is Florence2 fine-tuned or the base model? (I know that in LiMAC with Florence2, Florence2 would be fine-tuned, but it's not fully clear whether this is the case for Florence2 alone).
4. How, exactly, is "end-to-end accuracy" calculated? Why would predicting "wait"s rather than "input-text"s increase overall accuracy? I understand that you try to explain this on lines 369–403, but a clear definition of the accuracy calculation would make this paragraph make more sense.

Minor issues:
- On line 118, "that treat" should probably be "that they treat"
- On line 142, it's a bit weird that the full architecture link links back to the current section

**Questions:**

Please see the weaknesses section, parts 1–4.

5. You mention that LiMAC "address[es] the computational constraints inherent to smartphone". Have you tried running it on a smartphone? If so, how well did it go?

---

> ### Author Response · Authors · 2024-11-19
>
> We would like to thank the reviewer for their review and address their concerns below.
>
> 1. Positional Encodings for Nesting:
>
> The use of positional embeddings allows us to capture the relative positioning of UI elements, helping AcT understand which elements are nearby or grouped together. Without positional embeddings, the order of elements on the screen would be less meaningful. In our approach, nesting information is implicitly encoded in the UI icons themselves, as these icons often have overlapping pixels. By combining this spatial information with the positional embeddings, AcT is able to effectively distinguish between nested and non-nested elements. We attempted to incorporate depth embeddings into the nested structure, combining these with positional embeddings to precisely represent the nesting of UI elements. However, this integration did not yield any noticeable improvement in performance. We hope this clarifies the role of positional embeddings in handling nested UI elements.
>
> 2. Distinct Action Types:
>
> *We have updated the paper to reflect this more clearly, and have added a link to the relevant appendix in Section 3.3.*
>
> The distinct action types are outlined in Appendix A of the paper, specifically Table 5. Though there are only 10 distinct action types in AndroidControl, and 8 in AitW, there are a combined 11 distinct actions. The distinct actions are: (1) open-app, (2) click, (3) long-press, (4) input-text, (5-8) scroll (right/left/up/down), (9) navigate-home, (10) navigate-back, and (11) wait. open-app, wait, and long-press do not feature in AitW, while navigate-home does not feature in AndroidControl.
>
> 3. Florence Fine-Tuned or Not?
>
> Yes, it is fine-tuned. The Florence2 baseline refers to the same Florence2 model used within our LiMAC framework.  Practically, we fine-tune Florence2 only once for each dataset, and we use it both as part of LiMAC as well as to compare against it.
>
> 4. End-to-End Accuracy:
>
> *We have improved the wording of this explanation in our paper, section 4.2.*
>
> Overall accuracy is dependent on both the action type and the action specifications. When predicting a “wait” action, if the action type is predicted correctly, this immediately yields a positive result for that timestep. However, when predicting actions with additional parameters, such as “input-text”, even if the action type is predicted correctly, the overall result may be negative if the VLM fails to fill the “text” input field correctly. Using an example to try and illustrate this, take an episode with 10 steps, where the correct action is “wait” for 5 steps and “input-text” for the other 5. Predicting “wait” for all 10 steps will yield an overall accuracy of 0.5. However, predicting “input-text” for all 10 steps will yield an accuracy of 0.5 * (probability of the VLM to give the correct text input). If this VLM accuracy is 80%, the overall accuracy will be 0.5 * 0.8 = 0.4.
>
> Minor Issues: Thank you for pointing these out! These have now been fixed.
>
> 5. Running on Smartphone:
>
> We have not yet explored deploying the model on smartphones. Instead, we focus on training and comparing agents in simulated environments, with an understanding that limited computational capacity is a real and key constraint in mobile devices. By leveraging open-source datasets, we aim to improve performance and efficiency in these settings. The smaller size of AcT and its faster response times, however, highlight its promising potential for smartphone applications in the future.

---

### Official Review · Reviewer_Wgue · 2024-11-03

**Soundness:** 3
**Presentation:** 3
**Contribution:** 3
**Rating:** 8
**Confidence:** 3

**Summary:**

This paper introduces LiMAC (Lightweight Multi-modal App Control), a new architecture designed for mobile app control that combines a lightweight transformer, AcT, with a fine-tuned vision-language model (VLM). The standout feature here is the gated architecture, which smartly assigns tasks: AcT handles basic interactions like clicks and scrolls, while the VLM is called upon only for text-based actions that require deeper language comprehension. The approach yields substantial improvements in both inference speed and accuracy on two mobile action datasets.

**Strengths:**

1. **Novel design.** The authors designed a lightweight module to predict the type of actions to be taken, and execute simple actions with this light-weight module directly. Leaving the VLM to solve complex tasks that involve text generation. This leads to both performance speed-up and better accuracy.

2. **Thorough evaluations.** I like how the authors compared using AcT/VLM for different tasks, clearly demonstrating the performance gain by adopting the current design, which makes sense to me.

3. **Good writing.** The paper is easy to follow.

**Weaknesses:**

1. **Limited Dataset and Tasks.** The authors used two datasets of relatively small size, this paper could benefit from larger-scale experiments and maybe real-world user studies.

2. Due to the limited data size, the proposed model may have additional difficulties in solving difficult tasks (which is where the mobile AI is needed to most, from my opinion). More studies/analysis on failure mode could make this paper better.

**Questions:**

See my suggestions in the weakness section, here are some of my questions:

1. How does the model handle dynamic UI elements or applications with frequently changing interfaces? Do you need to retrain a model for each software update?

2. What is the impact of the VLM size on the overall performance? Could one larger VLM learn to solve all the tasks?

3. How does the system handle errors or recovery from incorrect actions? Or safegaurding it from performing unwanted actions (for example send out user's passwords to someone else).

---

> ### Author Response · Authors · 2024-11-19
>
> We would like to thank the reviewer for their review and address their concerns below.
>
> Weaknesses:
> 1. Limited Datasets and Tasks:
>
> We appreciate the reviewer’s concern regarding the limited size of datasets in our study. While we agree that a larger dataset would provide valuable insights, our work focuses on open-source data, and gathering additional data presents significant (and out of scope) challenges. Specifically, collecting data through emulator interaction is difficult, because Android emulators typically do not offer reward functions to assess task success. AndroidWorld [1] is the only exception we are aware of, though it is limited to just 116 tasks. We are actively working to address this limitation, in future work, by solving the challenges associated with  Android emulators, which will allow for a wider and more diverse set of tasks. This limitation is mentioned in the conclusion of the paper.
>
> 2. More Analysis of Failure Modes:
>
> *We thank the reviewer for suggesting a deeper analysis of failure modes is a valuable suggestion. We have added such an analysis in the Appendix D3 (Figure 4).*
>
> Questions:
>
> 1. Handling of Dynamic Elements and Changing Interfaces:
>
> Regarding the handling of dynamic elements and shifting interfaces, retraining would be required when transitioning to completely different operating systems, such as from Android to iOS. However, within the same OS, we expect the accuracy drop across different versions to be minimal, having observed only slight performance variations between train and test sets, and with AitW for example containing a range of Android OS versions and phone models.
>
> 2. Impact of VLM Size:
>
> We believe that training larger Vision-Language Models (VLMs) could potentially improve performance. However, as suggested in the AndroidControl paper [2], achieving 95% accuracy may require up to 60 million high quality episodes, using the PaLM-S model, which may contain over 100B parameters. While scaling up models is crucial, it is also important to recognise the significant resources required to reach such levels of performance (and that is why believe novel research directions that do not focus simply on scale are valuable).
>
> 3. Error handling and Recovery:
>
> *We now briefly discuss and acknowledge this limitation in Sec 6.*
>
> In the current scope of our work, we focus primarily on the core functionalities and performance evaluation of LiMAC. While comprehensive mechanisms for error recovery are essential, they were not the primary focus. We recognise the importance of robust error handling and recovery processes, and we intend to address these in future work. We also agree that safeguarding personal data is a concern, and while the current version of LiMAC does not specifically address this, we have included this as a direction of future research in the conclusion.
>
> [1] Rawles et al. AndroidWorld: A Dynamic Benchmarking Environment for Autonomous Agents.
>
> [2] Li et al. On the Effects of Data Scale on Computer Control Agents.

---

> > ### Comment · Reviewer_Wgue · 2024-11-20
> >
> > Thanks for the detailed response, that answered most of my questions.

---

### Official Review · Reviewer_mpP2 · 2024-11-05

**Soundness:** 3
**Presentation:** 3
**Contribution:** 3
**Rating:** 6
**Confidence:** 4

**Summary:**

The paper introduces Lightweight Multi-modal App Control (LiMAC), a framework designed for efficient mobile app control by combining a small Action Transformer (AcT) and a fine-tuned vision-language model (VLM). LiMAC processes user goals and smartphone states, making real-time decisions through AcT for action types like clicking or scrolling. For complex tasks requiring text input, it leverages the VLM to generate appropriate content. Evaluation of LiMAC on AndroidControl and Android-in-the-Wild datasets, LiMAC shows superior accuracy and speed over traditional VLMs and foundation models like GPT-4o, achieving up to 42% higher action accuracy and reducing task completion time by 30-fold. This approach emphasizes efficient model use on resource-limited devices, while future improvements may incorporate reinforcement learning for enhanced performance.

**Strengths:**

- LiMAC combines a small Action Transformer (AcT) with a fine-tuned vision-language model (VLM). This hybrid approach is tailored to the computational constraints of mobile devices, achieving efficient and accurate control without relying on large, resource-intensive models. The AcT independently handles common actions, while the VLM is selectively employed for complex natural language tasks, optimizing both resource usage and response time.
- LiMAC’s modular structure supports the integration of different models for specialized tasks, such as using AcT for action type prediction and click targeting, while optionally substituting modules for specific needs (e.g., Qwen2-VL for text generation tasks). This flexibility enables developers to adapt LiMAC easily for varied app control tasks, optimizing specific components without overhauling the entire architecture​.
- The AcT component employs a contrastive objective to identify UI elements for click actions, using cosine similarity and a learnable temperature parameter. This approach is advantageous in handling class imbalance and dynamically varying UI elements across episodes. The use of contrastive learning allows AcT to focus on directional alignment in the embedding space, facilitating precise UI element targeting even in dense or complex layouts

**Weaknesses:**

-  Although the paper evaluates LiMAC on two datasets, both datasets are relatively specific to Android applications, potentially limiting the generalizability of results to other operating systems (e.g., iOS) or app control tasks with distinct interface designs.
- The paper does not provide an extensive scalability analysis of LiMAC’s architecture as task complexity or the number of available UI elements increases, which may impact its robustness in more complex or densely populated app environments.
- LiMAC operates within a fixed action space, which could restrict its adaptability to applications requiring more dynamic or unconventional actions not included in its current design. This might hinder its flexibility in expanding to novel app interaction scenarios.
- The evaluation is conducted on simulated datasets without testing LiMAC’s performance on actual mobile devices. This limits understanding of its practical usability, particularly concerning latency, responsiveness, and potential challenges from hardware constraints and real-world conditions.

**Questions:**

See Weaknesses.

---

> ### Author Response · Authors · 2024-11-19
>
> We would like to thank the reviewer for their review and reply to their concerns below.
>
> 1. Android Focus:
>
> We focus on Android because it offers two open-source datasets and has a significant share of the global market as a mobile phone OS. These factors make our findings highly relevant and broadly applicable. We also want to emphasise that our methodology is flexible and can be adapted to other operating systems, such as iOS, once similar datasets become available.
>
> 2. UI Element Scalability:
>
> *We have included a study in Appendix D4 (Figure 5) that examines how the number of UI elements affects the ratio of successful to unsuccessful action predictions.*
>
> In the evaluated datasets, the number of UI elements per observation varies from 1 to 290, with most observations having fewer than 100 UI elements. While it's possible for an observation to exceed this range, potentially leading to out-of-distribution issues, we believe that the current span adequately encompasses the bulk of practical scenarios encountered in app control. This ensures that our model is capable of managing a diverse range of tasks.
>
> 3. Fixed Action Space:
>
> *We have updated our description of the action space in Appendix A*
>
> While the action space is fixed, it is the same as the AitW and AndroidControl datasets (where it is also fixed) and aligns with prior work on app agents and emulators, such as SeeAct [1], AndroidWorld [2], AppAgent [3], MobileAgent [4], DigiRL [5]. These actions are grounded by how a human interacts with the phone (e.g., click, or swipe). This fixed action space covers the vast majority of actions available on Android devices, and we believe it provides a robust representation of real-world app control behaviour.
>
> 4. Simulated Data vs. Real-World Devices:
>
> We acknowledge the concern that the approach may face challenges in real-world settings. However, this is a broader issue in the field, not unique to our approach and requires significant amounts of work. By leveraging open-source datasets, we aim to improve performance and efficiency in the simulated setting as a foundation. The smaller size of AcT and its faster response times highlight its promising potential for on-device smartphone applications in the future.
>
> [1] Zheng et al. Gpt-4V(ision) is a Generalist Web Agent, if Grounded.
>
> [2] Rawles et al. Androidworld: A Dynamic Benchmarking Environment for Autonomous Agents.
>
> [3] Yang et al. Appagent: Multimodal Agents as Smartphone Users.
>
> [4] Wang et al. Mobile-agent: Autonomous Multi-modal Mobile Device Agent with Visual Perception.
>
> [5] Bai et al. DigiRL: Training In-The-Wild Device-Control Agents with Autonomous Reinforcement Learning.

---

### Meta-Review · Area_Chair_GxbY · 2024-12-16

**Metareview:**

The paper introduces Lightweight Multi-modal App Control (LiMAC), a lightweight and modular to take actions on mobile user interfaces. The framework consists of a small Action Transformer (AcT) which predicts which action (e.g. click, scroll, type)  needs to be taken, and, depending on this output, a fine-tuned VLM may be queried to help with more complex tasks, such as text input or opening an app. AcT leverages multi-modal embeddings for each UI element in the current screenshot to help predict the next action. Overall, the paper presents a focused contribution about making lightweight UI interaction agents. The experiments performed show improvements over existing methods both in terms of accuracy and inference cost. The interactive AI community will find insights from this paper useful towards building on-device systems.

Reviewers consistently mentioned that the paper is well-written and the experiments as well as ablations are sound. All reviewers agreed that using a light-weight action transformer for simpler actions, while offloading more complex actions to a fine-tuned VLM is a novel design choice and has useful practical applications such as on-device deployment.

Some reviewers (Wgue, mpP2, skyy) mentioned that the datasets used for evaluation are limited (specific to Android). and will benefit from online evaluations. Authors responded to these concerns by saying that mobile navigation datasets are limited to Android, and the infrastructure to run online evaluation is still lacking and previous works have only managed to run small-scale online evaluation studies. Instead, they stick to offline evaluations but utilise bigger training and evaluation datasets.

Additionally, as pointed out by one reviewer (Wgue), one of the limitations of the current work is the lack of focus or experiments about recovering from incorrect action (which is a very important skill for UI interaction). While this is left for future work, recovering from error (planning) might require bigger models and the paper will benefit from discussions around how their lightweight approach can be incorporated when a stronger bigger model is needed for more complex part of UI Navigation (which is reasoning / planning).

**Additional Comments On Reviewer Discussion:**

During the course of the rebuttal period, the authors addressed some concerns and provided more clarifications around novelty, experiment design and results:

1. Novelty: While the use of embeddings for individual UI elements has been explored before, the overall architecture of using a lightweight model for simpler decision making and offloading complex tasks to VLM is novel. Additionally, low level implementation differ (using hidden states of transformer rather than the embedding directly).

2. Clarifications: The authors provided various clarifications about the technical details of the approach which makes the paper stronger. Adding clarifications about position encodings, action types, metrics, etc in the main manuscript will improve clarity. Additional experiments / results during the rebuttal should also be added in the appendix.

3. Finally, I encourage authors to update the discussion section to include analysis of failure modes in the main paper.

---

### Decision · Program_Chairs · 2025-01-22

Accept (Spotlight)